# Study on Screening Parameter Optimization of Wet Sand and Gravel Particles Using the GWO-SVR Algorithm

Jiacheng Zhou [1,2], Libin Zhang [1,2], Longchao Cao [1], Zhen Wang [2], Hui Zhang [2], Min Shen [2], Zilong Wang [2] and Fang Liu [2,*]

1. Hubei Key Laboratory of Digital Textile Equipment, Wuhan Textile University, Wuhan 430073, China; zhoujiacheng0609@126.com (J.Z.)
2. School of Mechanical Engineering and Automation, Wuhan Textile University, Wuhan 430073, China
* Correspondence: 2010024@wtu.edu.cn; Tel.: +86-139-8606-7408

**Abstract:** The optimization of screening parameters will directly improve the screening performance of vibration screens, which has been a concern of the industry. In this work, the discrete element model of wet sand and gravel particles is established, and the vibration screening process is simulated using the discrete element method (DEM). The screening efficiency and time are used as evaluation indices, and the screening parameters including amplitude, vibration frequency, vibration direction angle, screen surface inclination, the long and short half-axis ratio of the track, feeding rate, and screen surface length are investigated. The results of an orthogonal experiment and range analysis show that the amplitude, screen surface inclination, and vibration frequency are significant factors affecting screening performance. Then, the support vector regression optimized with the grey wolf optimizer (GWO-SVR) algorithm is used to model the screening data. The screening model with excellent learning and prediction ability is obtained with the Gaussian kernel function setting. Moreover, the GWO-SVR algorithm is used to optimize the screening parameters, and the screening parameters with optimal screening efficiency and time are obtained. Furthermore, the effectiveness and reliability of the optimized model are verified using the discrete element calculation. The optimization strategy proposed in this work could provide guidance for the structural design of vibration screens and screening process optimization.

**Keywords:** screening parameters; discrete element method (DEM); support vector regression; grey wolf optimizer; screening efficiency and time

## 1. Introduction

With the advancement in urbanization, sand and gravel, being high-quality building materials, are widely used for the construction of buildings, roads, and bridges [1–4]. Technologies used to make wet sand and gravel are widely used in areas with sufficient water sources because of the good appearance of production and minimal dust created in the process [5]. Vibration screening of wet sand and gravel is an important link in the production process [6]. Therefore, the research on wet sand vibration screening becomes particularly critical.

Due to the poor working environment, complex screening situation, and limited technical conditions, the stability and efficiency of screening experiments are poor [7], which makes it difficult for experimental research to go deep. With the vigorous development of particle technology and computer science, the numerical simulation method has greatly improved the deficiencies and defects in screening experiments [8–10]. The screening research conducted using numerical simulation software not only reduces the cost but also can greatly improve work efficiency [11]. Dong presents a numerical study on the particle flow on a banana screen at the particle scale using the discrete element method (DEM) [12]. The mathematical models relating the looseness coefficient to time are established using the

least squares method by Li [13]. The effect of vibration intensity on the screening process is discussed, and the potential error induced in the analysis of a single factor on the screening performance is demonstrated [14]. Liu found that when the values for the inclination of discharge and increment of screen deck inclination are 10° and 5°, respectively, the banana screening process obtains good screening performance in the simulation [15].

The accuracy of the particle model will directly determine the accuracy of results in particle screening simulations [16–18]. At present, numerical models for geological technology applications have been studied by some scholars. The NLSSI methodology for application to nuclear facilities for both the design and beyond-design basis ground motions was proposed by Coleman [19]. A parametric study was conducted to assess the effectiveness of the SC mitigation technique by gradually increasing the extension of remediation in order to achieve a satisfactory lower level of permanent deformation [20]. A 3D finite element analysis framework was presented in an attempt to address a number of salient features associated with the seismic response of wharf-ground systems [21]. However, there are few reports on the modeling of wet sand and gravel particles, and the adhesion between particles is not clear. Therefore, it is necessary to study the wet sand particle.

Meanwhile, the research shows that screening parameters have a great influence on screening performance [22,23]. However, there is no direct correlation between screening parameters, which makes it difficult to describe the influence of various screening parameters on the screening results using an explicit mathematical model. With the development of computer science and technology, endless machine-learning methods are emerging [24–26]. These methods can accurately predict unknown data by modeling and analyzing the existing data [27] and can also be applied to the study of screening parameters. The novel application of non-linear regression modeling with support vector machines (SVMs) was used to map the sample space of the operating parameters and vibrating screen configuration by Li [28]. The nonlinear principal component of the vibration signal was extracted, and a machine-learning model was constructed using LS-SVM, which reduced the AR coefficient and improved the learning ability and speed of the model [29]. In addition, a hybrid MACO-GBDT algorithm based on ant colony optimization (ACO) was also proposed to optimize the sieving performance of the vibrating screen by Chen [30].

However, there are still some challenges in the application of machine learning methods for screening parameters. The neural network method requires a large number of training data samples [31]. Nevertheless, it is difficult to obtain a large amount of data between screening efficiency and screening time whether using simulations or experiments. The accuracy of a machine learning model will be greatly reduced when the amount of data is insufficient. The support vector machine method can predict data using a small data sample, but the selection and optimization of the internal parameters in the model is a new problem. The traditional particle swarm optimization (PSO) algorithm is slow and unstable when optimizing the internal parameters of SVM [32], and it is very easy to fall into a locally optimal solution in the optimization process, which means that the support vector machine method optimized with the algorithm will also have defects and deficiencies [33]. Therefore, finding a better screening parameter model is also an urgent problem to be solved.

The GWO-SVR algorithm is used to construct an association model to optimize the screening parameters of wet sand and gravel particles. The association model constructed using the GWO-SVR algorithm has a strong global search ability [34], a fast convergence rate, and high precision [35]. Meanwhile, the data sample required for the association model training is small [36], which is particularly suitable for screening process optimization where it is difficult to obtain a large number of screening data samples. The results prove that the GWO-SVR algorithm has obvious advantages over traditional algorithms (MACO-GBDT, PSO-SVR, et al.).

In this work, the different wet sand and gravel discrete element models are first established using parameter calibration. Then, the screening efficiency and screening

time under different screening parameters are obtained using discrete element simulation. The screening parameter model is constructed using support vector regression, which is optimized with the gray wolf algorithm, and the appropriate kernel function is used to improve learning and prediction ability. Moreover, the optimized screening parameter model is constructed, and the screening parameters with optimal screening efficiency and screening time are obtained. Furthermore, the feasibility and reliability of the optimization method are verified using the discrete element test.

## 2. Simulation of the Screening Process

### 2.1. Discrete Element Modeling of Wet Sand and Gravel Particles

The contact model of the particles is one of the key factors affecting the accuracy of discrete element simulation [37]. The Hertz–Mindlin model is the main contact model used in the study of particle motion [38], which can simulate the interaction between dry particles but cannot reflect the mutual adhesion between wet particles. However, the JKR contact model can make up for the deficiency of the Hertz–Mindlin contact model [39,40].

The surface adhesion between wet particles refers to the corresponding surface energy in the JKR contact model. For the convenience of description, the particles are replaced with spheres. The simplified model is shown in Figure 1. The contact radius between the two particles expands from $\alpha_1$ to $\alpha_2$ due to the effect of surface energy [41]. The bonding force between particles can be expressed as $W$ [42].

$$W = \gamma_1 + \gamma_2 - \gamma_{12} \tag{1}$$

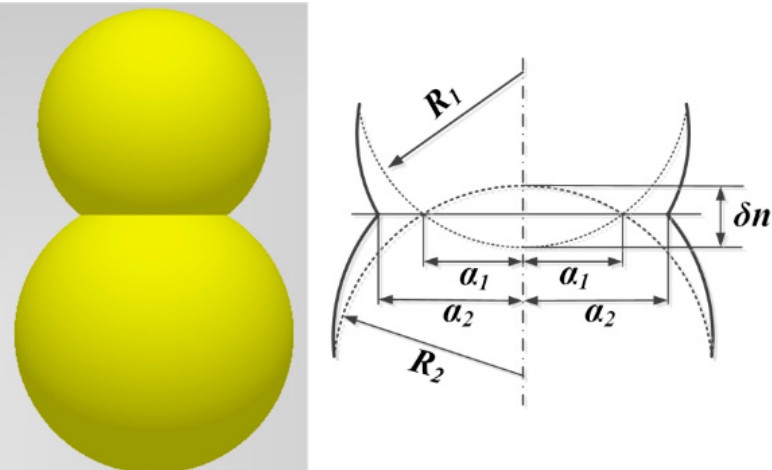

**Figure 1.** The JKR bonding model.

In Equation (1), $\gamma_1$ represents the surface energy of particle 1, $\gamma_2$ represents the surface energy of particle 2, and $\gamma_{12}$ denotes the boundary energy between particle 1 and 2. For the same particle type, the surface energies of different particles are the same, and the boundary energy between sand and gravel particles is 0. Then, $\gamma_{12} = 0$ and $\gamma_1 = \gamma_2 = \gamma$. Therefore, the cohesive force between the same particle type is $W = 2\gamma$. The normal elastic contact force $F_{JKR}$ and normal overlap $\delta$ of wet particles are shown in Equation (2) and Equation (3), respectively.

$$F_{JKR} = -2\sqrt{2\pi W E^* a_2^3} + \frac{4E^* a_2^3}{3R^*} \tag{2}$$

$$\text{ffi} = \frac{a_2^2}{R^*} - \sqrt{\frac{2\pi W a_2}{E^*}} \tag{3}$$

Substituting $W = 2\gamma$ into Equations (2) and (3), the normal elastic contact force $F_{JKR}$ and normal overlap $\delta$ can be expressed as:

$$F_{JKR} = -4\sqrt{\pi\gamma E^* a_2^3} + \frac{4E^* a_2^3}{3R^*} \tag{4}$$

$$\delta_n = \frac{a_2^2}{R^*} - 2\sqrt{\frac{\pi\gamma a_2}{E^*}} \tag{5}$$

where $\gamma$ is the surface energy of the particle, $E^*$ is the energy efficiency elastic modulus, $\alpha_2$ is the radius of the contact surface between the two particles in a collision, and $R^*$ is the equivalent contact radius. Next, there are:

$$\frac{1}{E^*} = \frac{1 - v_1^2}{E_1} + \frac{1 - v_2^2}{E_2} \tag{6}$$

$$\frac{1}{R^*} = \frac{1}{R_1} + \frac{1}{R_2} \tag{7}$$

In Equations (6) and (7), $E_1$, $v_1$, and $R_1$ are the elastic modulus, Poisson's ratio, and the radius of particle 1, respectively, and $E_2$, $v_2$, and $R_2$ represent the same physical values for particle 2 as particle 1, respectively. When the surface energy $\gamma$ of the particle is 0, the normal elastic contact force $F_{JKR}$ of the model can be simplified to the contact force $F_{Hert}$ in the Hertz model.

In addition to the ideal contact model, the particle shape is another important factor affecting the accuracy of discrete element simulation [43–45]. After jaw crushing and impact crushing of ores with large particle sizes [46,47], the main components in the particle groups are flat, triangular cone, and ellipsoid particles.

The three different shapes of wet sand and gravel particles are modeled, and the corresponding discrete element model is modeled using the filling ball method. Wet sand and gravel particle models with different shapes are 3D scanned and appropriately simplified. Then, the particle models are meshed using the FEM (finite element method). Three sizes of filled balls (2 mm, 1.5 mm, and 1 mm) are placed into the specified mesh nodes, and the construction of three wet sand and gravel particle models was then completed [48].

The modeling process is shown in Figure 2. The sizes of flat, triangular cone, and ellipsoid particles are 40–45 mm, 15–18 mm, and 10–12 mm, respectively. The flat, triangular cone and ellipsoid particles correspond to obstructing particles, difficult to screen particles, and easy to screen particles, respectively. With the extraction of particles from the vibration screen device for the component analysis, it is found that the quantity ratio of flat, triangular cone, and ellipsoid particles is about 1:72:97. Considering the size of the simulation model, the number of flat, triangular cone, and ellipsoid particles is set to 250, 17,500, and 25,000, respectively.

In previous studies, the wet bonding between particles and shapes was discussed in detail [49]. In order to ensure the validity of the particle model and improve the simulation accuracy, a calibration simulation and experiment on the wet sand and gravel particles were carried out. The angle of repose in the cylinder lifting test under different simulation parameter settings was optimized, and the set parameter combination closest to the experimental simulation was obtained. Figure 3 shows the calibration experiment setup for the wet particles.

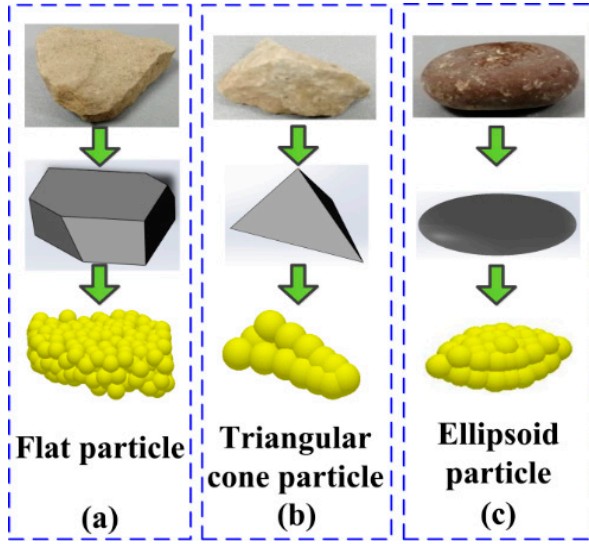

**Figure 2.** Discrete element model of wet sand and gravel particles with different shapes: (**a**) flat particle, (**b**) triangular cone particle, and (**c**) ellipsoid particle.

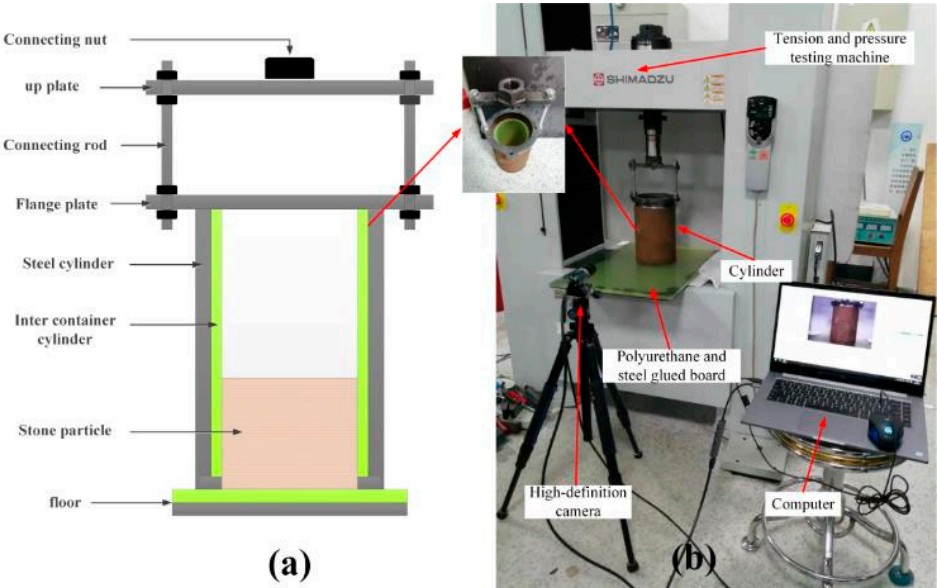

**Figure 3.** Experimental setup for the wet particle calibration: (**a**) schematic diagram of cylindrical device, and (**b**) experiment scene of cylinder lifting.

The simulation and experimental angle of repose in the three wet particles are shown in Figure 4. It is found that the surface energy between wet particles and the rolling friction coefficient between wet particles and the polyurethane material are the significant factors affecting the motion of particles with the parameter calibration of the three different shapes of the wet sand and gravel particles. For a comprehensive consideration of simulation accuracy and calculation efficiency, the surface energy and rolling friction coefficient corresponding to the optimal filling ball radius are shown in Table 1.

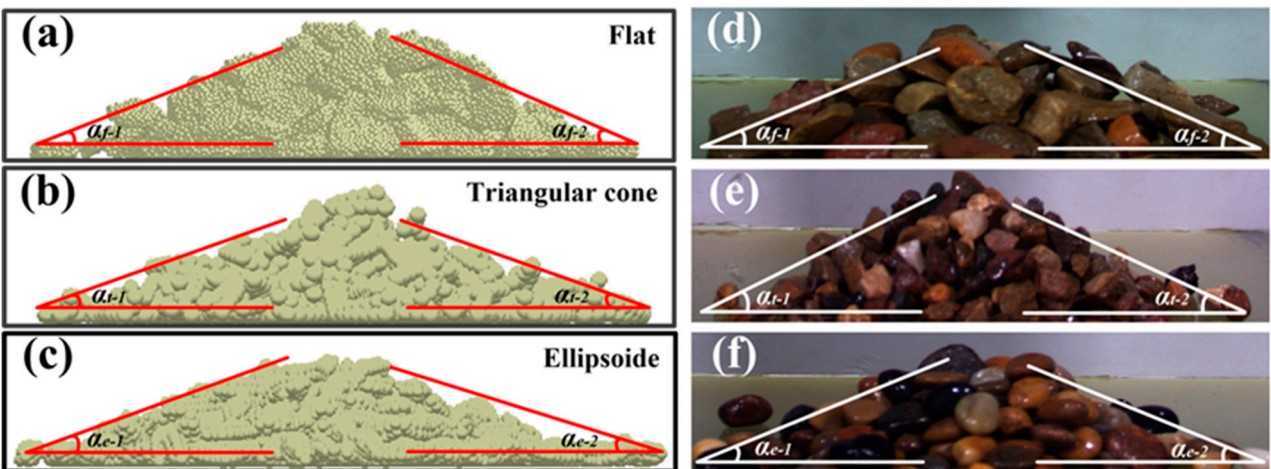

**Figure 4.** Simulation (**a**–**c**) and experiment (**d**–**f**) for cylinder lifting.

**Table 1.** Significant factors influencing the contact parameters of wet sand and gravel.

| Particle Type | Radius ofFilling Ball (mm) | Surface Energy (J·m$^{-2}$) | Rolling Friction Coefficient |
|---|---|---|---|
| Flat | 3.5 | 0.180206 | 0.032 |
| Triangular cone | 3 | 0.1907 | 0.029 |
| Ellipsoid | 1 | 0.2078 | 0.29 |

The parameters of the non-significant factors have the same value for the three wet sand and gravel particles, as shown in Table 2.

**Table 2.** Non-significant factors influencing the contact parameters of wet sand and gravel.

| Parameter | Value |
|---|---|
| Collision recovery coefficient between particle | 0.35 |
| Collision recovery coefficient between particle and polyurethane plate | 0.25 |
| Static friction coefficient between particle | 0.3 |
| Static friction coefficient between particle and polyurethane plate | 0.625 |
| Rolling friction coefficient between particle and polyurethane plate | 0.05 |

### 2.2. Screening Simulation Model

To improve the efficiency of simulation, the experimental model for the elliptical vibration screen [50,51] is simplified under the premise of meeting the requirements of the screening function. The simplified vibration screen in Figure 5 is mainly composed of a feed port, screen box, screen mesh, and bottom box. The geometric parameters in the vibration screen model are shown in Table 3. It is noteworthy that the screen box is designed to be semi-closed to present the particle splashing process. There will be a small number of particles moving out of the screen model during the simulation, which is consistent with the actual industrial production. The bottom box is divided into two parts: the receiving area and blanking area. The undersized fine particles are collected in the receiving area, and the oversized coarse particles are collected in the blanking area.

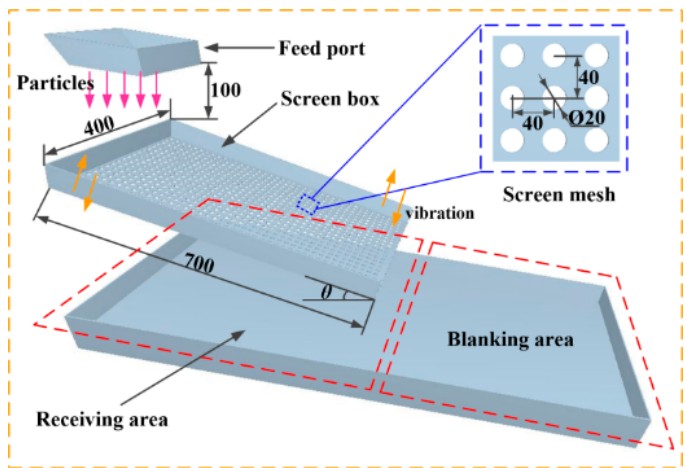

**Figure 5.** Discrete element model for the vibration screen (dimensions shown in mm).

**Table 3.** Geometric parameters in the vibration screen model.

| Parameters | Value (mm) |
|---|---|
| Screen length | 700 |
| Screen width | 400 |
| Screen thickness | 2 |
| Aperture size | 20 |
| Feeding height | 100 |
| Receiving area length | 700 |
| Blanking area length | 500 |

The parameters in the vibration screen model used for the discrete element simulation are shown in Table 4. Except for the screen mesh, which is made of polyurethane, the rest of the components are made of steel. With polyurethane material, the screen mesh has better wear resistance and longer service life [52].

**Table 4.** Material parameters in the vibration screen model and particles.

| Material | Components | Poisson's Ratio | Shear Modulus | Density |
|---|---|---|---|---|
| Stone | particles | 0.25 | 50 MPa | 2500 kg/m$^3$ |
| Steel | screen box | 0.27 | 79.92 GPa | 7850 kg/m$^3$ |
| Polyurethane | screen mesh | 0.43 | 500 MPa | 1100 kg/m$^3$ |

The force analysis of wet sand and gravel particles on the screen surface is shown in Figure 6. When the wet sand and gravel particles move on the vibration screen, the force equation when the particles are in contact with the screen is Equation (8).

$$\begin{cases} ma_x = mg\sin\theta - F_f \\ ma_y = F_n - mg\cos\theta \end{cases} \tag{8}$$

The accelerations in the X and Y directions of the wet particles are given by Equation (9):

$$\begin{cases} a_x = \frac{\Phi l}{2}\omega^2\cos\omega t\cos\alpha \\ a_y = \frac{\Phi l}{2}\omega^2\cos\omega t\sin\alpha \end{cases} \tag{9}$$

Let $D = \frac{\Phi l}{2}$, and then Equation (8) can be transformed into Equation (10).

$$\begin{cases} -mD\omega^2\cos\omega t\cos\alpha + mg\sin\theta = F_f \\ mD\omega^2\cos\omega t\sin\alpha + mg\cos\theta = F_n \end{cases} \tag{10}$$

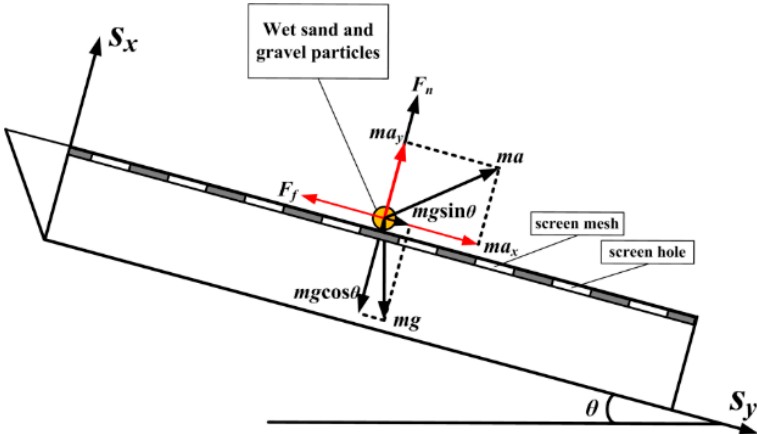

**Figure 6.** Force diagram showing the particles on the screen mesh.

In Equation (10), $m$ is mass of a single wet particle. $F_n$ and $F_f$ represent the normal force and friction force on the wet particles on the screen surface, respectively. $l$ represents the distance between point A on the screen surface and the wet particle. $D$ is the moving distance of the wet particle. When the wet particle leaves the screen surface, the normal force $F_n = 0$. Then, there is:

$$mD\omega^2cos\varphi_d sin\alpha + mgcos\theta = 0 \tag{11}$$

In Equation (11), $\varphi_d$ is the throwing start angle, which is the critical value for the vibration phase angle when the wet sand particles are thrown.

Three kinds of wet sand and gravel particles with different shapes and sizes fall from the feed port into the screen box and screen mesh area. The screen box and screen mesh move back and forth according to the elliptical track to throw up the mixed particles for screening. The discrete element simulation process for the vibrating screening of wet sand and gravel particles is shown in Figure 7. Three kinds of particles with different sizes are marked using different colors for convenience of observation. Flat particles, triangular cone particles, and ellipsoidal particles are marked as red, blue, and yellow, respectively. It can be seen from Figure 7 that the blue triangular cone particles and yellow ellipsoid particles have been thoroughly screened in the front section of the screen mesh. Fine particles passing through the screen mesh fall into the receiving area in the bottom screen box. The red flat particles fail to pass through the screen mesh and fall into the blanking area in the bottom screen box. The server used in the above simulation calculation has 512 G of RAM and 80 cores. Each simulation calculation time under the different screening parameters is about 160–200 h.

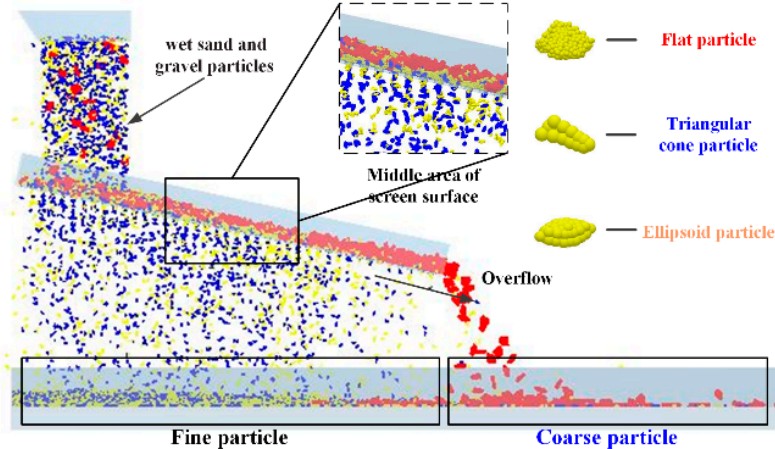

**Figure 7.** DEM simulation of the vibration screening process.

*2.3. Screening Performance Index: Screening Efficiency and Time*

The screening efficiency and time were used to evaluate the screening performance of the vibration screen.

Screening efficiency refers to the ratio of the actual mass of the product under the screen in the screening operation to the weight of the particles whose sizes are smaller than the size of the screen hole. Screening efficiency is an important index used to judge the quality of vibrating screening, which represents the degree of screening operation and the quality of screened products. The higher the screening efficiency, the better the screening process. The screening efficiency $\eta$ is shown in Equation (12) [53].

$$J = \frac{100 \times (a - b)}{a(100 - b)} \tag{12}$$

In Equation (12), *a* is the percentage content of the screened particles smaller than the size of the screen hole and *b* is the percentage content of the product particles on the screen smaller than the size of the screen hole.

In addition, screening time is an important index used to judge the throughput of vibrating screening, which represents the total amount of screening particles completed in unit time in the screening operation. The shorter the screening time, the higher the screening throughput and the higher the production efficiency. For a certain number of screened particles, the time from the beginning of the particle falling to the time when all particles leave the screen mesh is called the screening time. Considering the adhesion of the wet sand and gravel on the screen surface, the screening time *t* for the screening of wet sand and gravel particles is determined by the moment when the remaining particles on the screen weigh 5% of all the input materials after feeding ends.

## 3. Prediction of the Screening Performance

### 3.1. Support Vector Machine

Support vector machine (SVM) is one of the machine learning methods with super learning performance that was developed in recent years [54–56]. It can find the best compromise between the complexity of the model and the learning ability according to limited sample data, so as to obtain the best generalization ability [57]. Support vector machine has many unique advantages in solving small sample, nonlinear, and high-dimensional pattern recognition [58], and it can be extended to other machine learning problems such as function fitting [59].

### 3.2. The Grey Wolf Optimizer

The grey wolf optimizer (GWO) is an intelligent optimization algorithm proposed by Mirjalili in 2014 [60], which is an efficient search algorithm inspired by grey wolf hunting. Compared with the traditional particle swarm optimizer (PSO), the grey wolf optimizer (GWO) has a strong global search and convergence performance as well as few characteristic parameters and easy implementation [61,62].

Grey wolves are gregarious canines that are at the top of the food chain. As shown in the pyramid shape in the lower right corner of Figure 8, the grey wolf strictly abides by the hierarchy of social domination. In the pyramid, $\alpha$, $\beta$, $\delta$, and $\omega$ are located in the first to fourth levels of the wolf society. The wolves in the lower level of the social class must obey the orders of the wolves in the upper level of the social class. When the wolves in the upper social class are going senile or die, the wolves in the next level will become the best candidates to enter this group. GWO includes the steps of grey wolf society, tracking, encircling, and attacking prey. The grey wolf hunting process is shown on the left side of Figure 8.

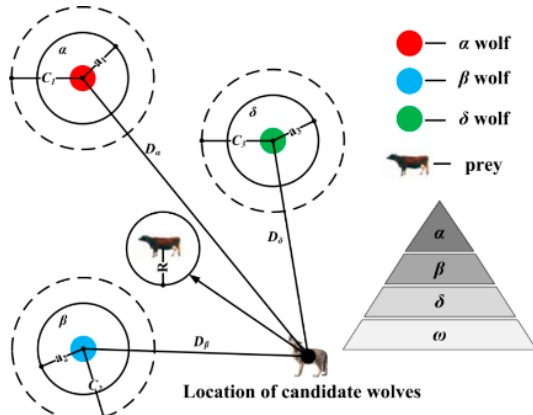

**Figure 8.** Schematic diagram showing the grey wolf hunting process.

Grey wolves have the ability to identify the location of potential prey (optimal solution), and the search process mainly depends on the $\alpha$, $\beta$, and $\delta$ wolf finish. However, the spatial characteristics of the solution in many problems are unknown, and the grey wolf is unable to determine the exact location of prey (optimal solution). In order to simulate the search behavior of a grey wolf (candidate solution), suppose $\alpha$, $\beta$, and $\delta$ have a strong ability to identify the location of potential prey. Therefore, the best three grey wolves ($\alpha$, $\beta$, and $\delta$) in the current race are retained during the whole recursive process, and then the location of other search agents is updated according to their location information. The mathematical model of this process is shown in Equation (13).

$$D_\alpha = C_1 \bigcirc X_\alpha - X, D_\beta = C_2 \bigcirc X_\beta - X, D_\delta = C_3 \bigcirc X_\delta - X \tag{13}$$

$$X_1 = X_\alpha - A_1 \bigcirc D_\alpha, X_2 = X_\beta - A_2 \bigcirc D_\beta, X_3 = X_\delta - A_3 \bigcirc D_\delta \tag{14}$$

$$X(t+1) = \frac{X_1 + X_2 + X_3}{3} \tag{15}$$

where $X_\alpha$, $X_\beta$, and $X_\beta$ represent the position vector of $\alpha$, $\beta$, and $\delta$ in the current population, respectively. $X$ represents the position vector of the grey wolf, and $D_\alpha$, $D_\beta$, and $D_\delta$ represent the distance between the current candidate grey wolf and the best three wolves, respectively. When $|A| > 1$, the grey wolves try to disperse in various areas and search for prey. When $|A| < 1$, the grey wolves will focus on searching for prey in a certain area.

It can be seen from Figure 8 that the position of the candidate solution finally falls within the random circle position defined by $\alpha$, $\beta$, and $\delta$. In general, $\alpha$, $\beta$, and $\delta$ need to first predict the approximate location of the prey (potential optimal solution), and then the other candidate wolves randomly update their location near the prey under the guidance of the current wolf of the optimal solution.

### 3.3. Construction of the Screening Parameter Prediction Model

At present, the GWO-SVR model is well applied to a variety of different prediction and detection projects. A prediction model for landslide displacement is established according to the variational mode decomposition (VMD) and support vector regression (SVR) optimized with the gray wolf optimizer (GWO-SVR), and the results indicate that the newly proposed model achieves a relatively good prediction accuracy with data decomposition and parameter optimization [63]. Owing to the serious interference from soil moisture content in the detection techniques, such as the X-ray fluorescence spectroscopy XRF method, a support vector regression SVR correction prediction model is proposed using the grey wolf optimization GWO algorithm, and the results show that the SVR nonlinear model has a better decision coefficient and smaller errors than the linear regression model [64]. In

order to detect the moisture content in green tea effectively and accurately, the dielectric technology combined with the VISSA-GWO-SVR model for nondestructive determination of the moisture content in tea is proposed, which will provide a promising tool for the moisture content detection of other agricultural products [65].

To further improve the learning and prediction capability of SVR, the kernel function $\delta$ and penalty coefficient $C$ in the SVR are optimized with the grey wolf algorithm [66], and the optimal screening parameters are determined for the screening data. The flow chart showing the screening prediction model established with GWO-SVR is shown in Figure 9.

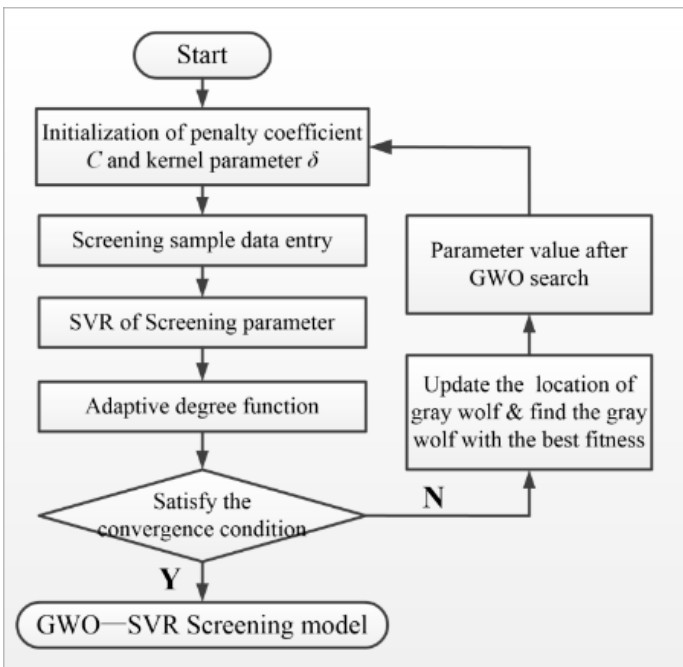

**Figure 9.** Workflow fir the GWO-SVR.

(i) The first step is to initialize the parameters in GWO and set the initial values of the penalty coefficient $C$ and kernel function $\delta$. The initial position of each wolf in population is determined so that the wolves with the best fitness can be selected more easily.

(ii) After calculating the training value and test value of each wolf in the training sample and test sample, the relative error value is then defined as the fitness function.

(iii) After comparing the fitness function value for the current wolf with the best wolf, the position of the current wolf is updated. Meanwhile, the synergy coefficient vector A and C are updated to help to find the position of the best wolf.

(iv) If the set convergence condition is not satisfied when the gray wolf algorithm is at the maximum number of iterations, the process will return to the second step for parameter re-optimization until the parameters that meet the convergence conditions are selected.

According to the relationship between screening efficiency, screening time, and the screening parameters, the optimization parameters and optimization objective expression are defined in Equation (16).

$$\begin{cases} \min \tau_e = \Sigma \left( 0.5 \times \frac{(Y_{e-train} - Y_e)^2}{N_{e-train}} + 0.5 \times \frac{(Y_{e-test} - Y_{e-t})^2}{N_{e-train}} \right) \\ min \tau_t = \Sigma \left( 0.5 \times \frac{(Y_{t-train} - Y_t)^2}{N_{t-train}} + 0.5 \times \frac{(Y_{t-test} - Y_{t-t})^2}{N_{t-train}} \right) \end{cases} \qquad (16)$$

where $min\tau_e$ and $min\tau_t$ represent the minimum value of the fitness function training and testing error in the GWO-SVR model for screening efficiency and screening time, respectively. The meanings of the various parameters in Equation (16) are as follows:

$Y_{e\text{-train}}$ and $Y_{t\text{-train}}$ are the training values of screening efficiency and screening time, respectively. $Y_e$ and $Y_t$ are the actual values of the training samples for screening efficiency and screening time, respectively.

$Y_{e\text{-test}}$ and $Y_{t\text{-test}}$ are the predicted values of the test samples for screening efficiency and screening time, respectively.

$Y_{e\text{-t}}$ and $Y_{t\text{-t}}$ are the actual values of test samples for screening efficiency and screening time, respectively.

$N_{e\text{-train}}$ and $N_{t\text{-train}}$ are the number of training samples for screening efficiency and screening time, respectively.

$N_{e\text{-test}}$ and $N_{t\text{-test}}$ are the number of test samples for screening efficiency and screening time, respectively.

The next step is to define the parameters in the screening model as the independent variable of *X*. The implicit function for screening efficiency $\eta$ and screening time *t* is shown in Equation (17). In Equation (26), the different screening parameters are the inputs of the screening model, and screening efficiency and screening time are the outputs of the screening model, respectively.

$$\begin{cases} \eta = f_1(X) \\ t = f_2(X) \end{cases} \tag{17}$$

where *X* is the vector of seven screening parameters in the elliptical vibrating screen model, as is shown in Equation (18).

$$X = [x_1, x_2, x_3, x_4, x_5, x_6, x_7]^{\mathrm{T}} \tag{18}$$

The meanings of parameters in Equation (27) are as follows: $x_1$ is the amplitude *a* (mm), $x_2$ is the vibration frequency *f* (Hz), $x_3$ is the vibration direction angle $\alpha$ (°), $x_4$ is the inclination angle of the screen surface $\theta$ (°), $x_5$ is the ratio of the long and short half axes of the track $b/a$, $x_6$ is the feeding rate *v* (m/s), and $x_7$ is the length of the screen surface *L* (mm).

*3.4. Orthogonal Experimental Table Design*

To explore the influence of different screening parameters on screening efficiency and screening time, orthogonal experiments on the screening parameters were designed. As mentioned above, there are seven main factors affecting screening performance: amplitude, vibration frequency, vibration direction angle, the inclination angle of the screen surface, the ratio of the long and short half axes of the track, feed rate, and the length of the screen surface. To study the influence range of each factor more comprehensively, five levels were taken for each factor. The factors and levels in the orthogonal experiment are shown in Table 5. The orthogonal table with seven factors and five levels is designed for research. For an amplitude of factor 1, when the actual amplitude exceeds 8 mm, particles will present excessive splash. So, the range in amplitude is 2~8 mm. For a vibration frequency of factor 2, the particles cannot be loosened quickly when the actual vibration frequency is too small. So, the range in the vibration frequency is 12~20 Hz. For the vibration direction angle of factor 3, the direction angle range of a common vibration screen is 20~70°, according to the design experience of a vibration screen. So, the range in the vibration direction angle in this experiment is 25~65°. For the inclination angle of factor 4, the small inclination angle in the screen surface leads to low particle flow, while the large inclination angle is unfavorable for full particle penetration through the screen. So, the inclination angle in the screen surface is chosen between 12 and 18°. For factor 5, the ratio of the long and short half axes of the track is 0.2~1. For the feeding rate, a too-fast feeding rate easily reduces the screening efficiency, and a too-slow feeding rate will reduce output. So, the range of the feeding rate in this experiment is 0.5~2.5 m/s. For factor 7, the range of the

screen length is 660~740 mm, based on the actual vibration screen, to scale down the model. The orthogonal experiment table for the screening parameters was designed using the Design-Expert11 software [67], and the simulations and analyses were completed using EDEM software [68]. Finally, the screening efficiency and screening time corresponding to each group of parameters were recorded.

**Table 5.** Factors and levels in the orthogonal experiment.

| Trial | Factor 1 | Factor 2 | Factor 3 | Factor 4 | Factor 5 | Factor 6 | Factor 7 |
|---|---|---|---|---|---|---|---|
| | $a$ (mm) | $f$ (Hz) | $\alpha$ (°) | $\theta$ (°) | $b/a$ | $v$ (m/s) | $L$ (mm) |
| 1 | 2 | 12 | 25 | 10 | 0.2 | 0.5 | 660 |
| 2 | 3 | 14 | 35 | 12 | 0.4 | 1.0 | 680 |
| 3 | 4 | 16 | 45 | 14 | 0.6 | 1.5 | 700 |
| 4 | 5 | 18 | 55 | 16 | 0.8 | 2.0 | 720 |
| 5 | 6 | 20 | 65 | 18 | 1.0 | 2.5 | 740 |

### 3.5. Significance Analysis for the Screening Parameters

An analysis of the range is one of the commonly used analysis methods in orthogonal experiments, which is intuitive and easy to understand. The greater the range, the greater the influence of a parameter on the index. According to the orthogonal experimental results for the designed screening parameters, the importance of the above seven screening parameters is reordered. The results show that the order of importance for the factors affecting screening efficiency is vibration frequency > inclination > amplitude > long short half-axis ratio > direction angle > feed rate > screen length. The order of importance for the factors affecting screening time is inclination > vibration frequency > amplitude > feeding rate > long short half-axis ratio > direction angle > screen length. It can be seen that the vibration frequency, inclination, and amplitude have a great impact on screening efficiency and time.

### 3.6. Selection of the Kernel Function

In the development and application of SVM, the use of the kernel function makes it easier for linear SVM to be extended to nonlinear SVM. The kernel function is the soul of SVM and also determines its performance [69,70]. The GWO-SVR model is affected by the random position of hunting wolves in the hunting process of the grey wolf algorithm, which produces slightly different results each time. To make the prediction results more stable and to further improve the learning ability of the GWO-SVR screening model, the kernel functions and kernel parameters of the appropriate sample data should be selected [71]. The common kernel functions in SVM are shown in Table 6 [72].

**Table 6.** Common kernel function types in SVM.

| Linear Kernel Function: $K(x_i \cdot x_j) = x_i \cdot x_j$ | Polynomial Kernel Function $K(x_i \cdot x_j) = (x_i \cdot x_j + 1)^d$ |
|---|---|
| Sigmoid kernel function: $K(x_i \cdot x_j) = \tanh(k x_i \cdot x_j - \delta)$ | Gaussian kernel function: $K(x_i \cdot x_j) = e^{\frac{\lVert x_i \cdot x_j \rVert^2}{2\delta^2}}$ |

To ensure that the screening model has good learning and generalization ability at the same time, the values of $\delta$ in the Gaussian kernel function and Sigmoid kernel function are set as {0.02:0.1:0.5, 1, 4, 8, 12, 16, 20}, and the values of order $d$ are set as {1, 2, 3}. To avoid over-fitting, a smaller soft interval is used. The value of the penalty coefficient $C$ will be controlled in a small range. The value of the penalty coefficient $C$ is 12, and the value of parameters $\varepsilon$ and $\lambda$ are 0.01 and $1 \times 10^{-8}$, respectively. The maximum number of iteration steps is 500.

According to the data obtained from the screening orthogonal experiment, the screening parameter model is established using the different types of kernel functions in Table 6.

Comparing the operation time and convergence for different types of kernel functions, it is found that the sigmoid function did not reach convergence after 500 iteration steps, and the polynomial kernel function has a long operation time. Meanwhile, the linear function and the Gaussian kernel function complete the operation and converge in a faster time. The iterative process of the GWO-SVR screening model for screening efficiency under the linear kernel function and radial basis kernel function is shown in Figure 10a,b. It can be seen from the iteration process of the linear kernel function and Gaussian kernel function that both kernel functions converge in the iteration process of 200 steps (Figure 11).

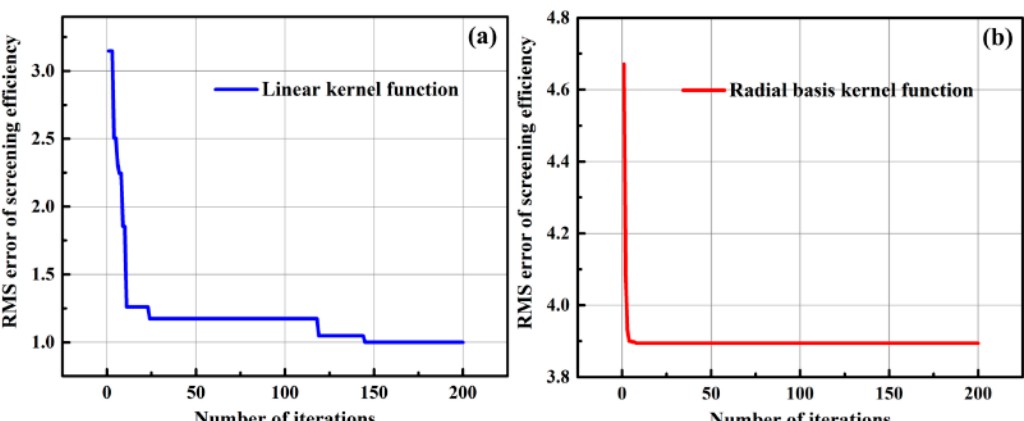

**Figure 10.** Iterative process of screening efficiency under two kernel functions: (**a**) liner kernel function, and (**b**) radial basis kernel function.

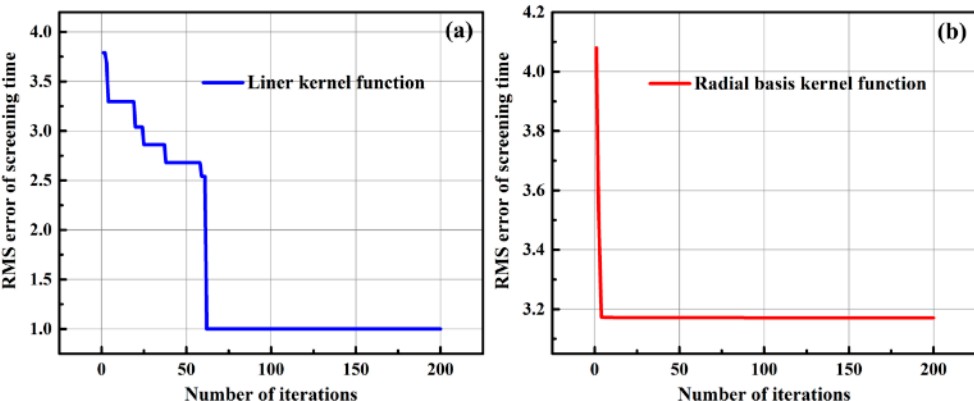

**Figure 11.** The iterative process for screening time under two kernel functions: (**a**) liner kernel function, and (**b**) radial basis kernel function.

The *RMSE* (Root Mean Square Error), *MAE* (Mean Absolute Error), and $R^2$ (R-Square) are used as the evaluation indicators for the performance of the GWO-SVR screening model [58]. The expression of *RMSE*, *MAE*, and $R^2$ are shown in Equation (19).

$$
\begin{cases}
RMSE(X,h) = \sqrt{\dfrac{1}{m}\sum\limits_{i=1}^{m}(h(x_i) - y_i)^2} \\
MAE = \dfrac{1}{m}\sum\limits_{i=1}^{m} |\, y_i - x_i \,| \\
R^2 = 1 - \dfrac{\sum_{i=1}^{m}(\hat{y}_i - y_i)^2}{\sum_{i=1}^{m}(\overline{y}_i - y_i)^2}
\end{cases}
\tag{19}
$$

where $m$ is the number of samples and $y_i$ and $\hat{y}_i$ are actual value and screening time learning and prediction performance under linear kernel function and Gaussian kernel function, respectively, as shown in Table 7.

**Table 7.** Performance of the GWO-SVR screening model under two kernel functions.

| Evaluating Indicator | Screening Efficiency $\eta$(%) | | Screening Time $t$(s) | |
|---|---|---|---|---|
| **Learning Objectives** | **Linear** | **Gaussian** | **Linear** | **Gaussian** |
| *RMSE* | 4.882 | 5.773 | 0.6708 | 0.6283 |
| *MAE* | 0.0832 | 0.0707 | 0.6322 | 0.5816 |
| $R^2$ | 0.7194 | 0.6465 | 0.5758 | 0.5163 |
| Running time (min) | 369.30 | 101.47 | 156.85 | 73.82 |

The normalization of MSE [73] and RMSM [74] are commonly used as criteria for measuring the prediction results of machine learning models. MAE [75] can better reflect the actual situation of prediction value errors. Therefore, in this work, MAE is used as the evaluation indicator for the performance of the GWO-SVR screening model instead of MSE.

Comparing the evaluation indices for the linear kernel function and Gaussian kernel function [76], as shown in Table 7, the prediction ability of the screening model under the linear kernel function is slightly stronger, but the difference between them is not significant. However, comparing the operation time of the screening model under the two kernel functions, it is found that the Gaussian kernel function takes significantly less time. Meanwhile, comparing the convergence rate of the two kernel functions, it is found that the number of iterations in the screening model is fewer, and the convergence rate is faster under the Gaussian kernel function for screening efficiency and screening time. Therefore, the Gaussian kernel function with fewer iterations and faster convergence is used as the kernel function of the GWO-SVR screening model in subsequent analyses.

### 3.7. Prediction Accuracy of the Screening Model

There are 300 groups of test data in the orthogonal experiment on the screening parameters. The first 220 groups of data are used as the test samples for the training group, and the last 80 groups as the test samples for the prediction group. It takes seven screening parameters as input, and the screening efficiency and screening time are the output. The training results of the GWO-SVR screening model on screening efficiency and screening time for the first 220 groups of test samples are shown in Figures 12 and 13. It can be seen from the two figures that the error between the experimental results and prediction results is small for both screening efficiency and screening time, which indicates that the learning and generalization ability of the GWO-SVR model has been improved using the Gaussian kernel function.

The prediction results of the GWO-SVR screening model on screening efficiency and screening time for the last 80 groups of test samples are shown in Figures 14 and 15. It can be seen from the two figures that the error between the experimental results and prediction results is small for both screening efficiency and screening time, which indicates that the prediction ability of the GWO-SVR model has been also improved using the Gaussian kernel function.

The above results show that GWO-SVR optimized with the Gaussian kernel function provides a reasonable calculation model for the complex and nonlinear mixed multidimensional space characteristics of the screening parameters. After testing the training group and prediction group for screening efficiency and screening time, it is found that this GWO-SVR screening parameter model has good learning, generalization, and prediction ability. It provides a reasonable mathematical model for the subsequent optimization of the screening parameters.

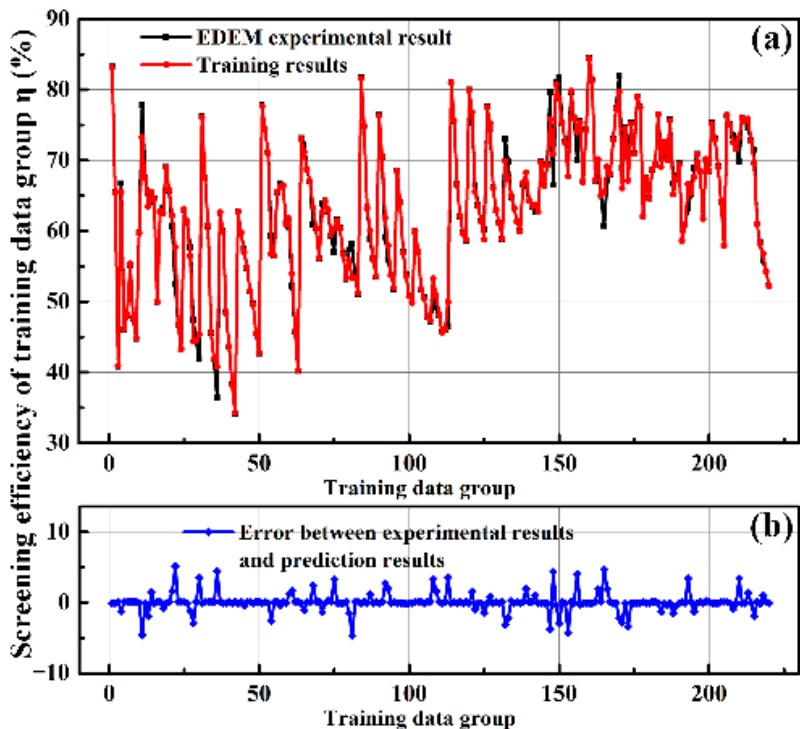

**Figure 12.** Comparison of screening efficiency between the training and experiment. (**a**) training results, and (**b**) error results.

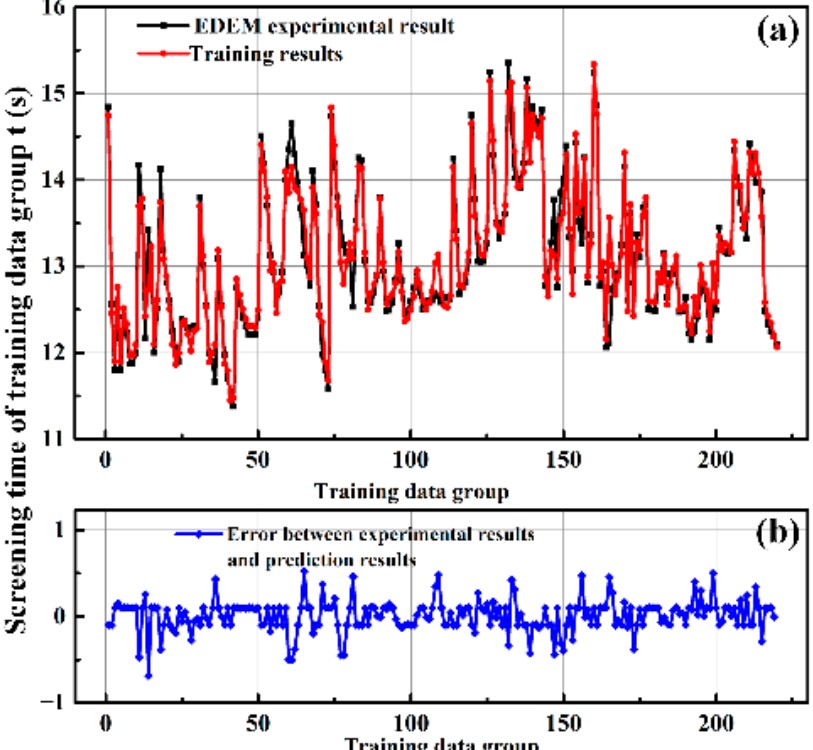

**Figure 13.** Comparison of screening time between the training and experiment. (**a**) training results, and (**b**) error results.

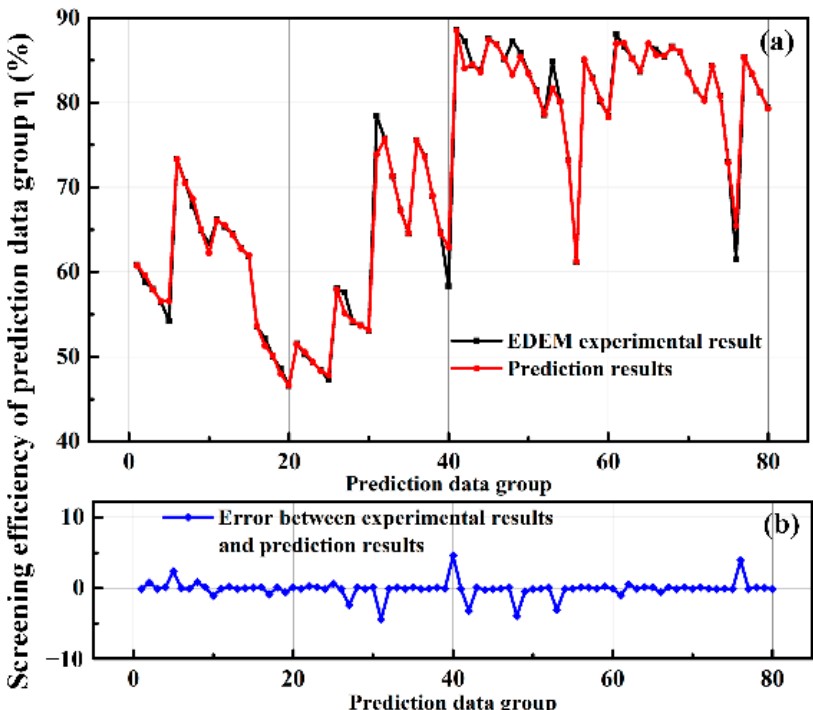

**Figure 14.** Comparison of screening efficiency between the prediction and experiment. (**a**) prediction results, and (**b**) error results.

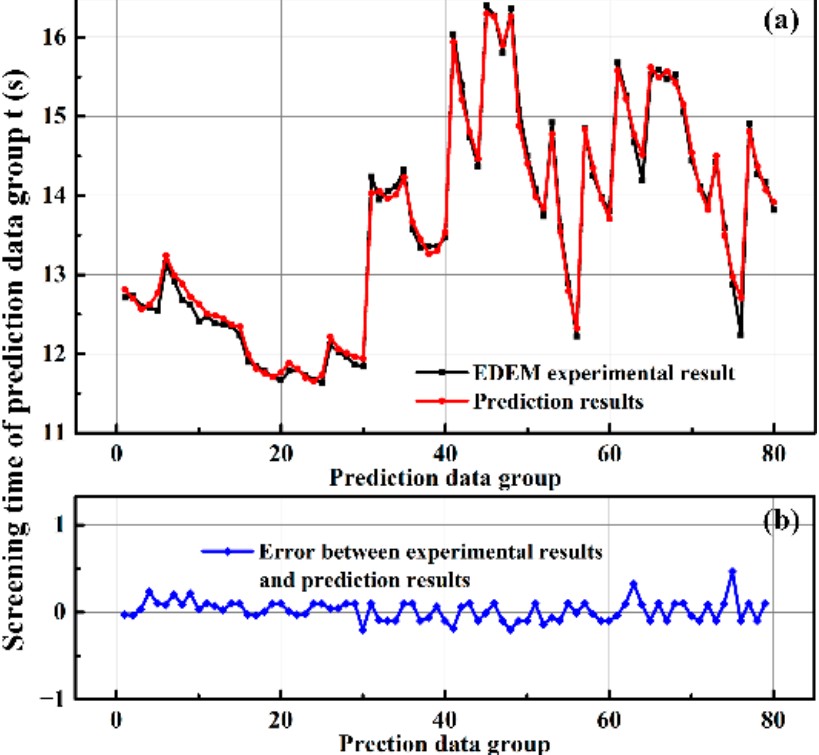

**Figure 15.** Comparison of screening time between the prediction and experiment. (**a**) prediction results, and (**b**) error results.

## 4. Optimization of the Screening Performance

The fundamental purpose of studying the screening parameters of a vibration screen is to improve the screening efficiency of the vibration screen and shorten its screening time so as to comprehensively improve the screening capacity of the vibration screen and

bring practical benefits to the industry. Therefore, the optimization design of screening parameters can provide great theoretical support for the reasonable design of the screen structure and the appropriate adjustment of the production process [77,78].

### 4.1. Construction of the Screening Parameter Optimization Model

The mathematical model for parameter optimization using the GWO-SVR screening model can be shown by Equation (20).

$$\begin{cases} max_{X \in D}\{f_1(X), f_2(X)\} \\ D = \{X \mid g_i(X) \leq 0, h_j(X) = 0; i = 1, 2, \ldots, m; j = 1, 2, \ldots, p\} \end{cases} \quad (20)$$

where $f_1(X)$ and $f_2(X)$ are the objective functions in the screening efficiency and screening time model, $g_i(X)$ and $h_j(X)$ are constraint functions, $D$ is the constraint region, and $X$ is the $n$-dimensional vector to be optimized. $f_1(X), f_2(X), g_i(X)$, and $h_j(X)$ are nonlinear functions. The GWO-SVR screening model is a highly nonlinear problem.

According to Equation (26), the GWO-SVR-optimized screening parameters are shown in Equation (21).

$$\begin{cases} max \; \eta = f_1(x_1, x_2, x_3, x_4, x_5, x_6, x_7) \\ min \; t = f_2(x_1, x_2, x_3, x_4, x_5, x_6, x_7) \end{cases} \quad (21)$$

The maximum values of screening efficiency and screening time under the combination of seven screening parameters are sought. The feasible range of the screening parameters is determined according to practical production experience. The range of screening parameters of the GWO-SVR-optimized screening model is shown in Equation (22).

$$s.t. \begin{cases} 2 \leq x_1 \leq 5, \; 10 \leq x_2 \leq 20, \; 25 \leq x_3 \leq 65, \\ 12 \leq x_4 \leq 20, \; 0.2 \leq x_5 \leq 0.8, \; 1 \leq x_6 \leq 2.5, \\ 640 \leq x_7 \leq 740, \end{cases} \quad (22)$$

The grey wolf algorithm is used to optimize the screening parameters of the GWO-SVR screening model. The training data come from the orthogonal test of the screening parameters. The Gaussian kernel function is used as the kernel function in SVR, and the minimum value of the error between the test data and prediction data is taken as the fitness function. The maximum number of iterations in the optimization process of screening efficiency and screening time is 500 steps. According to the training and learning results of the GWO-SVR screening model, the penalty coefficient $C_\eta$ is 49.394 and $\delta_\eta$ is 0.5756 in the optimal screening efficiency model. The penalty coefficient $C_t$ is 11.476 and $\delta_t$ is 0.2577 in the optimal screening time model.

### 4.2. Optimization Process and Results of the Screening Parameters

To compare the difference between grey wolf optimization and traditional particle swarm optimization for support vector machine, the optimization process of two algorithms for screening efficiency and screening time is shown in Figure 16.

It can be seen from Figure 16a that the optimization of GWO-SVR is nearly stable when the number of iterations in the screening efficiency optimization process is close to 400. However, the optimization of PSO-SVR still seems to have a changing trend at the 500th iteration step. At the 500th iteration step, the optimization results of screening efficiency in GWO-SVR and PSO-SVR are 88.37% and 85.86%, respectively. Meanwhile, it can be seen from Figure 16b that the optimization of GWO-SVR is nearly stable when the number of iterations in the screening time optimization process is close to 450. At the 500th iteration step, the optimization results of screening time for GWO-SVR and PSO-SVR are 11.83 s and 12.02 s, respectively. The convergence rate of the GWO-SVR model is faster than that of the PSO-SVR model in each iteration step in the optimization processes of screening efficiency and time. Overall, the screening efficiency and time of the GWO-SVR model are superior to those of the PSO-SVR model in terms of both convergence speed and optimization results.

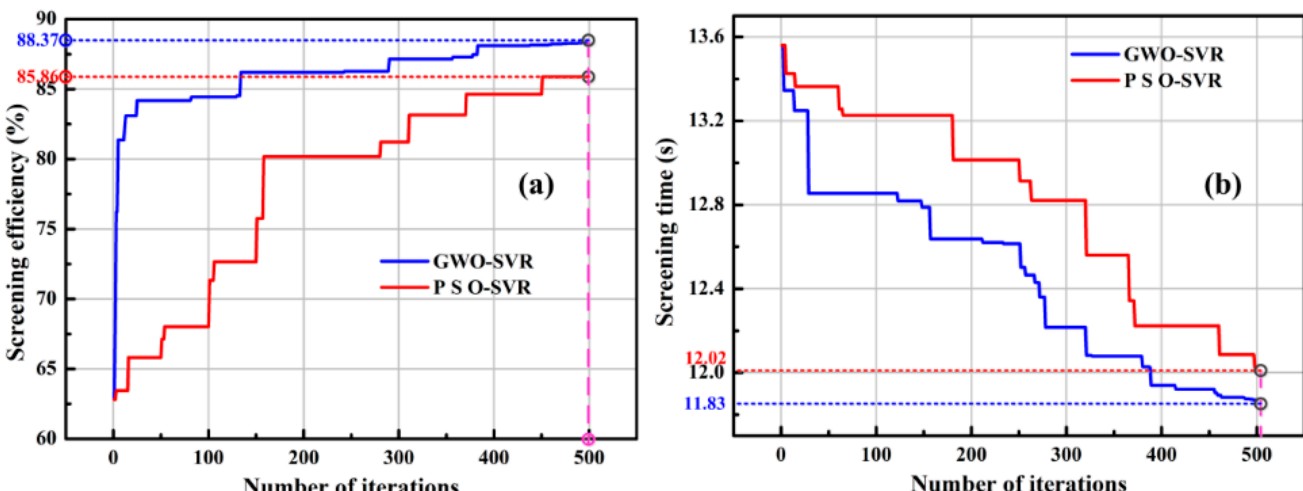

**Figure 16.** The optimization process for screening efficiency and time with GWO-SVR and PSO-SVR: (**a**) screening efficiency and (**b**) screening time.

Meanwhile, according to relevant research [64], for similar datasets (with fewer data samples and higher data dimensions), the prediction model constructed with GWO-SVR has better fitting performance compared to the ENR, Lasso, RR, and SVR algorithms. Compared to PSO-SVR, GA-SVR, and SA-SVR, GWO-SVR has stronger global search capabilities and convergence details [79].

Table 8 shows the screening parameters corresponding to the maximum screening efficiency and minimum screening time in the GWO-SVR screening optimization model.

**Table 8.** Screening parameters verified with EDEM.

| Optimization Objectives | $a$ (mm) | $f$ (Hz) | $\alpha$ (°) | $\theta$ (°) | $b/a$ | $v$ (m/s) | $L$ (mm) | Optimal Value |
|---|---|---|---|---|---|---|---|---|
| Screening efficiency $\eta$(%) | 3.0 | 17.1 | 45 | 12 | 0.35 | 1.4 | 660.0 | 88.37% |
| Screening time $t$(s) | 4.0 | 20.1 | 45 | 15.5 | 0.42 | 1.7 | 683.0 | 11.83 s |

*4.3. Verification of the Optimization Results*

To further verify the accuracy of the GWO-SVR screening optimization model, the screening parameters corresponding to the maximum screening efficiency and minimum screening time were tested using EDEM simulation.

As shown in Figure 17, the EDEM screening results show that the maximum screening efficiency is 83.24% and the minimum screening time is 12.24 s. Meanwhile, the maximum screening efficiency and minimum screening time predicted with the GWO-SVR screening optimization model under the same screening parameters are 88.37% and 11.83 s, respectively. The error of the maximum screening efficiency between predicted value and test value is 5.81%, while the error of the minimum screening time between predicted value and test value is 3.11%, which indicates that the errors in screening efficiency and time are within an acceptable range. Thus, the GWO-SVR screening optimization model is considered to be accurate and reliable. This provides a reference for the GWO-SVR algorithm in the screening optimization process.

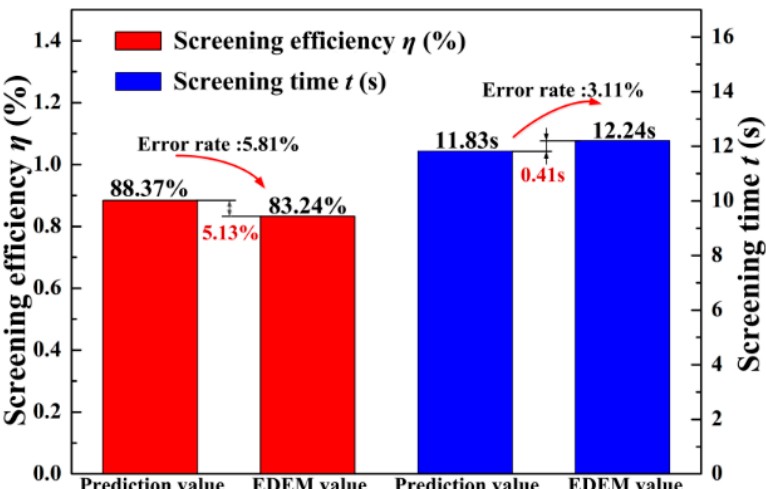

**Figure 17.** Value of prediction and EDEM of screening efficiency and screening time.

## 5. Conclusions

To explore the influence of different screening parameters on screening efficiency and screening time, the GWO-SVR algorithm was used to establish the screening parameter model. Seven screening parameters including the amplitude, vibration frequency, vibration direction angle, screening surface inclination, the long and short half-axis ratio of the track, feeding rate, and screen surface length were used as the influence factors of the screening performance. Based on the results, we draw the following conclusions:

(1) The discrete element model of wet sand and gravel particle screening was established first. The important factors affecting the screening process were obtained using an orthogonal experiment and range analysis. The results show that the amplitude, the screen surface inclination, and the vibration frequency are significant factors affecting screening efficiency and screening time.

(2) Then, the screening parameter model for screening efficiency and screening time based on the GWO-SVR algorithm was established. The learning and prediction ability of the screening parameter model is improved with the Gaussian kernel function. By comparing the prediction values and error in the training group and prediction group, it can be found that the GWO-SVR screening model has excellent learning and prediction ability for screening efficiency and screening time data. The error is within the acceptable range, which indicates the reliability of the GWO-SVR screening model.

(3) Furthermore, the optimal screening parameter model was constructed with the GWO-SVR algorithm, and the screening parameters with optimal screening efficiency and time were obtained. The maximum screening efficiency is 83.24%, while the minimum screening time is 12.24 s. Meanwhile, comparing the GWO-SVR algorithm with the PSO-SVR algorithm, it is found that the screening efficiency and time of the GWO-SVR model are superior to that of the PSO-SVR model in terms of both convergence speed and optimization results.

(4) Moreover, the screening parameters were used as input in EDEM to calculate the corresponding screening efficiency and screening time. We found that the calculated values are very close to the predicted values using the GWO-SVR algorithm. The above verification results prove the effectiveness and reliability of the optimization model.

The optimization method using the GWO-SVR algorithm proposed in this work provides guidance and reference for the subsequent structural design of vibration screens and also has great potential to improve the production quality and efficiency of the wet sand and gravel vibration screening industry.

**Author Contributions:** J.Z. and L.Z. designed and conducted the simulation and experiment and wrote the first draft of the article. L.C. and Z.W. (Zilong Wang) conducted the experimental analyses. H.Z. and M.S. contributed to the systematic review of the literature and data retrieval and extraction. Z.W. (Zhen Wang) conceived and analyzed the data and drafted the article. F.L. and Z.W. (Zhen Wang) funded this research and provided all the experimental devices. All authors have read and agreed to the published version of the manuscript.

**Funding:** This research work is funded by the National Natural Science Foundation of China (NSFC) under grant numbers 62204178, 51775388, 11872048, and 52105446; the Knowledge Innovation Program of Wuhan-Shuguang Project under grant number 2022010801020252; the Hubei Provincial Education Department Innovation Team Project under grant number T2022020; the Natural Science Foundation of Hubei Province under grant number 2022CFB995; and the Guidance Project of Science and Technology Research Program of Hubei Provincial Department of Education under grant number B2021107.

**Data Availability Statement:** The data used to support the findings of this study are available from the corresponding author upon request.

**Acknowledgments:** The authors would like to acknowledge the computer resources provided by Huazhong University of Science and Technology, and sincerely appreciate all the researchers involved for their meticulous efforts in this work.

**Conflicts of Interest:** The authors declare no conflict of interest.

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
