# Peer review of "Study on Screening Parameter Optimization of Wet Sand and Gravel Particles Using the GWO-SVR Algorithm"

_processes, doi:10.3390/pr11041283_

Round 1

Reviewer 1 Report

The paper investigates an interesting topic such as the screening parameter optimization of wet sand and gravel particles based on GWO-SVR algorithm. The methodology is pertinent and English is also good. However, there are several issues to be considered.

Introduction.

The novelty needs to be explained in detail to support the originality of the paper and its contribution to the research.

Is the GWO-SVR algorithm a novelty of the paper? If not, why the authors chose to use it? this needs to be explained in the introduction.

In addition, this part:

"In this work, the different wet sand and gravel discrete element models are firstly established by parameter calibration. Then, the screening efficiency and screening time under different screening parameters are obtained in discrete element simulation. The screening parameters model is constructed by support vector regression which is optimized by gray wolf algorithm, and the appropriate kernel function is adopted to improve learning and prediction ability. Moreover, the optimized screening parameters model is constructed, and the screening parameters with optimal screening efficiency and screening time are obtained. Furthermore, the feasibility and reliability of the optimization method are verified by discrete element test." is more pertinet in the methodology section.

Section 2

The authors wrothe that the three different shapes of wet sand and gravel particles are modeled and the corresponding discrete element model is modeled by filling ball method. However, it is not clear what the mentioned method is. The authors need to refer to it.

The test in Figure 5 is difficult to be read.

Section 3.1

Are these formulations original? If not, this section needs to be reduced.

Section 3.4

This part is difficult to be followed: "For the inclination angle of factor 4, the inclination angle of screen surface is too small to facilitate the downward flow of particles, while the inclination angle of the screen surface is too large to cause particles fall fast, which reduce the screening efficiency". please rewrite or reorganize in different sentences.

Section 3.7

Figures 13-16 have captions that are very difficult to be read. please refine them 

Section 4.3

The authors wrote that both errors are within 10%, which shows that the GWO-SVR screening optimization model is accurate and reliable. This does not seem convincing, unless supported properly with literature.

Author Response

Response to Reviewer1 # Comments

Reviewer1:

Comments and Suggestions to the Authors:The paper investigates an interesting topic such as the screening parameter optimization of wet sand and gravel particles based on GWO-SVR algorithm. The methodology is pertinent and English is also good. However, there are several issues to be considered. 1. Introduction. The novelty needs to be explained in detail to support the originality of the paper and its contribution to the research. Is the GWO-SVR algorithm a novelty of the paper? If not, why the authors chose to use it? This needs to be explained in the introduction.

Answer: Thanks for your suggestions. The novelty and advantages of the GWO-SVR algorithm has been explained in the introduction. The explanation of novelty and advantages is as follows:

In this work, the GWO-SVR algorithm is adopted to construct an association model to optimize screening parameters of wet sand and gravel particles. The association model constructed by GWO-SVR algorithm has strong global search ability [1], fast convergence rate, and high precision [2]. Meanwhile, the data sample required for the association model training is small [3], which is particularly suitable for screening process optimization where it is difficult to obtain a large number of screening data samples. The results prove that GWO-SVR algorithm has obvious advantages over traditional algorithms (MACO-GBDT, PSO-SVR, et al). 

The explanation of novelty and advantages are marked by red fonts to make them more prominent in the revised manuscript.

[1] J. Sun, Y.N. Mo, Y. Chen, N. Yang, Y. Tang, Detection of moisture content of tomato leaves based on dielectric properties and IRIV-GWO-SVR algorithm, T. ASABE 34(4) (2018):188-195. [2] Z. Yang, Y. Wang, C. Kong, Remaining useful life prediction of lithium-ion batteries based on a mixture of ensemble empirical mode decomposition and GWO-SVR model, IEEE T. Instrum. Meas. 70 (2021): 2517011.[3] A.L. Balogun, F. Rezaie, Q.B. Pham, L. Gigovie, S. Drobnjak. Y.A. Aina, M. Panahi, S.T. Yekeen, S. Lee, Spatial prediction of landslide susceptibility in western Serbia using hybrid support vector regression (SVR) with GWO, BAT and COA algorithms, Geosci. Front. 12 (2021): 101104. 2. In addition, this part: "In this work, the different wet sand and gravel discrete element models are firstly established by parameter calibration. Then, the screening efficiency and screening time under different screening parameters are obtained in discrete element simulation. The screening parameters model is constructed by support vector regression which is optimized by gray wolf algorithm, and the appropriate kernel function is adopted to improve learning and prediction ability. Moreover, the optimized screening parameters model is constructed, and the screening parameters with optimal screening efficiency and screening time are obtained. Furthermore, the feasibility and reliability of the optimization method are verified by discrete element test." is more pertinent in the methodology section.

Answer: Thanks for your constructive comments and suggestions. The above content is still retained in the introduction because there is no methodology section in this manuscript.

 3. The authors wrote that the three different shapes of wet sand and gravel particles are modeled and the corresponding discrete element model is modeled by filling ball method. However, it is not clear what the mentioned method is. The authors need to refer to it. Answer: Thanks for your constructive comments and suggestions. Wet sand and gravel particle models with different shapes are 3D scanned and appropriately simplified. Then, the particle models are mesh by FEM (finite element method). Three sizes of filled balls (2mm, 1.5mm, and 1mm) are placed into the specified mesh nodes, and the construction of three wet sand and gravel particle models has been completed [41].The above contents have been modified and marked by red fonts in the revised manuscript.[41] J.C. Zhou, L.B. Zhang, L.C. Cao, J.J. Tang, K.M. Mao, Study on the dynamics characteristics and screening performance of the disc spring vibration screen, J. Low Freq. Noise V. A. 67 (2023): 1-16. 4. The test in Figure 5 is difficult to be read.Answer: Thanks for your constructive comments. Figure 5 is a simplified DEM simulation model for wet sand and gravel particles screening separation. All explanatory text in the picture has been enlarged to facilitate better reading. Meanwhile, the movements of particles and screen box have been marked.

The above contents have been modified in Figure 5. And caption of Figure 5 has been marked by red fonts in the revised manuscript.

Figure 5. Discrete element model of vibration screen (dimension in mm).

5. Are Section 3.1 formulations original? If not, this section needs to be reduced.Answer: Thanks for your constructive comments and suggestions.Section 3.1 has been reduced to only retain an introduction to Support Vector Machine (SVM). The above contents have been modified in the revised manuscript. 6. This part is difficult to be followed in Section 3.4: "For the inclination angle of factor 4, the inclination angle of screen surface is too small to facilitate the downward flow of particles, while the inclination angle of the screen surface is too large to cause particles fall fast, which reduce the screening efficiency". Please rewrite or reorganize in different sentences.Answer: Thanks for your suggestions. For the inclination angle of factor 4, the small inclination angle of screen surface leads to low particle flow while the large inclination angle is unfavorable for full particle penetration through the screen. So the inclination angle of screen surface is chosen between 12~18°.The above contents have been modified and added and marked by red fonts in the revised manuscript. 7. Figures 13-16 have captions that are very difficult to be read in Section 3.7. Please refine them.Answer: Thanks for your constructive comments and suggestions.The captions of Figures 13-16 have been modified in Section 3.7, and modified captions can directly reflect the content in the figures.The above contents have been modified in the revised manuscript. 8. The authors wrote that both errors are within 10%, which shows that the GWO-SVR screening optimization model is accurate and reliable. This does not seem convincing, unless supported properly with literature in Section 4.3.Answer: Thanks for your constructive comments and suggestions. To verify the reliability of the GWO-SVR algorithm, the prediction has been imported into EDEM for further test. The error of the maximum screening efficiency between predicted value and test value is 5.81% while that of the minimum screening time between predicted value and test value is only 3.11%, which indicates that the errors of screening efficiency and time are within an acceptable range. Thus, the GWO-SVR screening optimization model is considered to be accurate and reliable.The above contents have been modified and marked by red fonts in the revised manuscript.

Reviewer 2 Report

The Manuscript entitled :Study on screening parameter optimization of wet sand and gravel particles based on GWO-SVR algorithm is very interesting scientific work. 

The Article is prepared for publishing however it benefits from revision following recommendations:

- Please avoid sets of references such as [1-4]. Please clarify why refer to all of these works not just one.

- Please describe more attempts made in case of prediction of similar problem with machine learning algorithms (not only SVR)

- Please provide literature survey what previously was these algorithms succesfully used for. (Grey Wolf optimization and SVR) to show the motivation why these exactly approach was taken.

- page 15. Why not use Normalised MSE and instead of two similar Errors RMSE and MSE maybe other different performance parameter?

- figures 14 -16 can be wider. It will be easier to see the charts. now both lanes are almost at the same place.

-It would be beneficial to discuss the results obtained in this work with other similar works in this field. It will prove if the model is accurate/better than other models or it is not significantly different that others.

- Conclusion should be seperated.

Author Response

Response to Reviewer2 # Comments

Reviewer2:

Comments to the Author

The Manuscript entitled: Study on screening parameter optimization of wet sand and gravel particles based on GWO-SVR algorithm is very interesting scientific work.

Answer: Thanks for your comments.

Comments for the author:

The Article is prepared for publishing however it benefits from revision following recommendations:

1. Please avoid sets of references such as [1-4]. Please clarify why refer to all of these works not just one.Answer: Thanks for your constructive comments and suggestions.

As mentioned at beginning of the introduction, application of sand and gravel in various aspects is introduced. The applications of sand and gravel in different aspects are referred to different references. Therefore, the sets of references such as [1-4] are kept in this manuscript.

 2. Please describe more attempts made in case of prediction of similar problem with machine learning algorithms (not only SVR).Answer: Thanks for your constructive comments and suggestions.Nonlinear principal component of vibration signal was extracted, and machine learning model is constructed by LS-SVM, which reduces the AR coefficient and improves learning ability and speed of model [1]. In addition, a hybrid MACO-GBDT algorithm based on the Ant Colony Optimization (ACO) was proposed for optimizing the sieving performance of vibrating screen by Chen [2].  The above contents have been modified and marked in the introduction part of revised manuscript.[1] B. Chen, D. Huang, F. Zhang, The Modeling Method of a Vibrating Screen Efficiency Prediction Based on KPCA and LS-SVM, Int. J. Pattern Recogn. 33 (2019): 19500009.[2] Z.Q. Chen, Z.F. Li, H.H. Xia, X. Tong, Performance optimization of the elliptically vibrating screen with a hybrid MACO-GBDT algorithm, Particuology. 56 (2021) 193-206.

  1. Please provide literature survey what previously was these algorithms successfully used for. (Grey Wolf optimization and SVR) to show the motivation why these exactly approach was taken.

Answer: Thanks for your constructive comments and suggestions.At present, GWO-SVR model is well applied to a variety of different prediction and detection projects. A prediction model of landslide displacement is established according to the variational mode decomposition (VMD) and support vector regression (SVR) optimized by gray wolf optimizer (GWO-SVR),which indicates that the newly proposed model achieves a relatively good prediction accuracy with data decomposition and parameter optimization [1]. Owing to the serious interference of soil moisture content in the detection techniques such as X-ray fluorescence spectroscopy XRF method a support vector regression SVR correction prediction model is proposed based on the grey wolf optimization GWO algorithm, and the results show that the SVR nonlinear model has a better decision coefficient and smaller errors than the linear regression model [2]. In order to detect moisture content in green tea effectively and accurately, dielectric technology combined with VISSA-GWO-SVR model for nondestructive determination of the moisture content in tea is proposed, which will provide a promising tool for the moisture content detection of other agricultural products [3].

The above contents have been added and marked by red fonts in the revised manuscript.

[1] C.H. Wang, W. Guo, Prediction of Landslide Displacement Based on the Variational Mode Decomposition and GWO-SVR Model, Sustainability 15(6) (2023): 5470.

[2] Y. Chen, C. Zhang, C.Y. Xiao, X.L. Zhao, Y.X. Shi, H. Yang, Z.Y. Liu, S.H. Li, Study on Prediction Model of Soil Cadmium Content Moisture Content Correction Based on GWO-SVR, Acta Opt. Sin. 40(10) (2020): 1030002.

[3] J. Sun, Y. Tian, X.H. Wu, C.X. Bai,B. Lu, Nondestructive detection for moisture content in green tea based on dielectric properties and VISSA-GWO-SVR algorithm, J. Food Process. Pres. 44(5) (2020): e14421.

  1. In page 15, why not use normalised MSE and instead of two similar Errors RMSE and MSE maybe other different performance parameter?

Answer: Thanks for your constructive comments and suggestions.Both normalization MSE [1] and RMSM [2] are commonly used as criteria for measuring the prediction of machine learning models. However, MAE [3] can better reflect the actual situation of prediction value errors. Therefore, MAE is adopted as the evaluation indicators for the performance of the GWO-SVR screening model instead of MSE in this work.

The above contents have been modified in Table 7 and marked by red fonts in revised manuscript.

[1] M. Panabi, A. Gayen, H.R. Pourghasemi, F. Rezaie, S. Lee, Spatial prediction of landslide susceptibility using hybrid support vector regression (SVR) and the adaptive neuro-fuzzy inference system (ANFIS) with various metaheuristic algorithms, Sci. Total Environ. 741(1) (2020): 139937.

[2] H.S. Dhiman, D. Dipankar, J.M. Guerrero, Hybrid machine intelligent SVR variants for wind forecasting and ramp events, Renew. Sust. Energ. Rev. 108 (2019): 369-379.

[3] C.Q. Luo, B. Keshtagar, S.P. Zhu, X.P. Niu, EMCS-SVR: Hybrid efficient and accurate enhanced simulation approach coupled with adaptive SVR for structural reliability analysis, Comput. Method. Appl. M. 400 (2022): 115499.

  1. Figures 13 -16 can be wider. It will be easier to see the charts. Now both lanes are almost at the same place.

Answer: Thanks for your constructive comments and suggestions.

The length-width ratios of figures 13-16 have been adjusted to make them look easier. The adjusted figures are shown below:

Figure 13. Comparison of screening efficiency between training and experiment

Figure 14. Comparison of screening time between training and experiment.

Figure 15. Comparison of screening efficiency between prediction and experiment.

Figure 16. Comparison of screening time between prediction and experiment.

The above contents have been modified and added and marked by red fonts in the revised manuscript.

  1. It would be beneficial to discuss the results obtained in this work with other similar works in this field. It will prove if the model is accurate/better than other models or it is not significantly different others.

Answer: Thanks for your constructive comments and suggestions. According to relevant research [1], for similar datasets (with fewer data samples and higher data dimensions), the prediction model constructed by GWO-SVR has better fitting performance compared to ENR, Lasso, RR, and SVR algorithms. Moreover, GWO-SVR has stronger global search capabilities and convergence details compared to PSO-SVR, GA-SVR, and SA-SVR [2].The above contents have been modified and added and marked by red fonts in the revised manuscript.

[1] S.L. Cong, J. Sun, H.P. Mao, X.H. Wu, W. Pei, X.D. Zhang, Non-destructive detection for mold colonies in rice based on hyperspectra and GWO-SVR, J. Sci. Food Agr. 98(4) (2017): 1453-1459.

[2] Y. Chen, C. Zhang, C.Y. Xiao, X.L. Zhao, Y.X. Shi, H. Yang, Z.Y. Liu, S.H. Li, Study on Prediction Model of Soil Cadmium Content Moisture Content Correction Based on GWO-SVR, Acta Opt. Sin. 40(10) (2020): 1030002.

  1. Conclusion should be separated.

Answer: Thanks for your constructive comments and suggestions.

The revised conclusions are as follows:

To explore the influence of different screening parameters on screening efficiency and screening time, the GWO-SVR algorithm is adopted to establish the screening parameter model. Seven screening parameters including the amplitude, vibration frequency, vibration direction angle, screening surface inclination, long and short half axis ratio of track, feeding rate and screen surface length are employed as the influence factors of screening performance. It can draw the following conclusions.

(1) The discrete element model of wet sand and gravel particle screening is established first. The important factors affecting the screening process are obtained through orthogonal experiment and range analysis. The results show that the amplitude, the screen surface inclination, and the vibration frequency are significant factors affecting screening efficiency and screening time.

(2) Then, the screening parameters model of screening efficiency and screening time based on GWO-SVR algorithm is established. The learning and prediction ability of screening parameter model is improved through Gaussian kernel function. By comparing the prediction values and error in the training group and prediction group, it can be found that the GWO-SVR screening model has excellent learning and prediction ability for screening efficiency and screening time data. The error is within the acceptable range, which indicates the reliability of the GWO-SVR screening model.

(3) Furthermore, the optimal screening parameters model is constructed by GWO-SVR algorithm, and the screening parameters with optimal screening efficiency and time are obtained. The maximum screening efficiency is 83.24% while the minimum screening time is 12.24s. Meanwhile, comparing GWO-SVR algorithm with PSO-SVR algorithm, it is found that the screening efficiency and time of the GWO-SVR model is superior to that of the PSO-SVR model in terms of both convergence speed and optimization results.

(4) Moreover, the screening parameters are used as input in EDEM to calculate the corresponding screening efficiency and screening time, we found that the calculated values are very close to the predicted values by GWO-SVR algorithm. The above verification results prove the effectiveness and reliability of the optimization model.

The optimization method of GWO-SVR algorithm proposed in this work provides guidance and reference for the subsequent structural design of vibration screen, and is also of great significance to improve the production quality and efficiency of wet sand and gravel vibration screening industry.

The above contents have been modified and marked by red fonts in the revised manuscript.

Round 2

Reviewer 1 Report

I consider that the reviews are fine and improved the paper. However, I suggest that the introduction should contain a discussion on numerical model of geotchnical applications. Please refer to:

Coleman JL, Bolisetti C, Whittaker AS (2016) Time-domain soil-structure interaction analysis of nuclear facilities, Nuclear Engineering and Design 298 (2016) 264–270.

D. Forcellini, A.M. Tarantino, Assessment of stone columns as a mitigation technique of liquefaction-induced effects during italian earthquakes (May 2012), Scientific World Journal, Article ID 216278, 201, 2014.

Su L, Lu J, Elgamal A, Arulmoli AK (2017) Seismic performance of a pile-supported wharf: three dimensional fnite element simulation. Soil Dyn Earthq Eng 95:167–179

Author Response

Answer: Thanks for your constructive comments and suggestions.

At present, the numerical models of geological technology applications have been studied by some scholars. The NLSSI methodology for application to nuclear facilities for both design and beyond-design basis ground motions is proposed by Coleman [19]. A parametric study was conducted to assess the effectiveness of SC mitigation technique by gradually increasing the extension of remediation, in order to achieve a satisfactory lower level of permanent deformation [20]. A 3D finite element analysis framework was presented in an attempt to address a number of salient features associated with the seismic response of wharf-ground systems [21].

The above contents have been added and marked by red fonts in the revised manuscript.

[19] J.L. Coleman, C. Bolisetti, A.S. Whittaker, Time-domain soil-structure interaction analysis of nuclear facilities, Nucl. Eng. DES 298 (2016): 264-270.

[20] D. Forcellini, A.M. Tarantino, Assessment of stone columns as a mitigation technique of liquefaction-induced effects during italian earthquakes (May 2012), Sci. World J 201 (2016): 216278.

[21] L. Su, A. Elgamal, A.K. Arulmoli, Seismic performance of a pile-supported wharf: three dimensional finite element methods.  Soil Dyn. Earthq Eng. 95 (2017): 167-179.    

Reviewer 2 Report

Manuscript can be published

Author Response

Thanks for your comments.